# A Low-Loss, 77 GHz, 8 × 8 Microstrip Butler Matrix on a High-Purity Fused-Silica (HPFS) Glass Substrate

**DOI:** 10.3390/s23031418

**Published:** 2023-01-27

**Authors:** Ronak Sakhiya, Sazzadur Chowdhury

**Affiliations:** Department of Electrical and Computer Engineering, University of Windsor, Windsor, ON N9B 3P4, Canada

**Keywords:** Butler matrix, passive microstrip beamformer, automotive radar, microfabrication, 3D FEM simulations

## Abstract

A low-loss, compact, ultra-thin, passive, 77 GHz, 8 × 8 microstrip Butler matrix on a 200 μm thick high-purity fused-silica (HPFS) glass substrate embedded in 0.8 μm thick patterned gold conducting layers was developed for low power automotive radars. The first-of-its-kind, HPFS, glass-based Butler matrix comprised 12 hybrid couplers, 16 crossovers, and 8 phase shifters in a footprint area of 19.1 mm × 26.6 mm. The device and the corresponding building blocks were designed and optimized using 3D electromagnetic finite element method (FEM) simulations using the Advanced Design System (ADS) from Keysight™ Technologies. Due to the very-low-loss tangent of the HPFS glass substrate (0.0005 @77 GHz) compared to other common substrate materials and rigorous design optimization, the return loss and isolation of the input ports are both below −20 dB, respectively, as verified by 3D FEM simulations. Due to the absence of any published data on a 77 GHz 8 × 8 Butler matrix, the design was validated by developing a 4 × 4 version of the Butler matrix using the same building blocks and comparing the 3D simulation results in ADS with results published elsewhere that showed that the developed Butler matrix offers lower insertion loss in a 10% smaller footprint area. A low-cost microfabrication method has been developed to fabricate the devices using a standard lift-off process. A scaled version of the device can be used for 5G beamforming applications.

## 1. Introduction

Automotive radars have become the key enabling technology for adaptive cruise control (ACC), autonomous driving, and collision mitigation (CM) applications for vehicles. Early automotive radars were designed to operate at 24 GHz center frequency. However, the narrow bandwidth of such radars is not suitable to achieving the necessary range and velocity accuracies. Their large size makes the system bulkier as well. Consequently, automotive manufacturers focused on developing automotive radars operating at 76–81 GHz to improve the range and velocity accuracies at a lower cost and with a smaller form factor while following the regulations and recommendations of the European Telecommunications Standard Institute (ETSI) and Federal Communications Commission (FCC) [1,2]. Additionally, as millimeter waves at high frequencies undergo very little absorption in human tissues due to their lower penetration through the human skin [3], the 76–81 GHz is safer compared to 24 GHz.

In any communication system, it is necessary to adjust the antenna beams in the appropriate direction so that the electromagnetic signals are transmitted and received by the end users with minimum signal loss. Modern communication systems consist of multibeam array antennas which are typically backed by radio frequency (RF) beamformers to achieve the target of beam steering with wide angle coverage [4]. Various types of passive RF beamforming techniques, such as Butler matrix [5], Rotman lens [6,7], Blass matrix [8], and Nolen matrix [9,10], have been reported. Of all these, the advantages of the Butler matrix make it an ideal candidate in terms of size, manufacturing costs, bandwidth, reliability, and reciprocity; therefore, it is widely used in several applications, such as Internet of Things (IoT), Wi-Fi, base stations, satellite communications, and automotive radars [11,12,13,14,15,16,17,18,19].

These applications are based on three types of radars—short-range radar (SRR), mid-range radar (MRR), and long-range radar (LRR). In Figure 1, the red beam depicts the LRR, blue beam depicts the MRR, and the green beam depicts the SRR. Typically, the specifications of the SRR, MRR, and LRR are as per Table 1.

An investigation by the authors showed that microfabricated microstrip Butler matrices operating at 77 GHz can be used in automotive radars to meet or enhance the target applications by providing a lower cost, more easily fabricated, more easily integrated, and lower in system complexity solution. Figure 2 shows the block diagram of a 77 GHz automotive radar that includes an 8 × 8 Butler matrix as the beamforming engine.

Both 4 × 4 and 8 × 8 are common Butler matrix configurations [11,12,13,14,15,16,17,18,19]. For applications where wider beams are necessary, a 4 × 4 configuration is preferred, and 8 × 8 Butler matrices are preferred for narrower beam applications. The physical size of a Butler matrix depends on the operating frequency and the choice of the substrate material. In automotive radars, 4 × 4 and 8 × 8 Butler matrices can be used for SRR, MRR, and LRR applications.

An excellent comprehensive analysis of Butler-matrix design variants that include bi-layer, tri-layer, and quad-layer geometries is available in [4]. For the bilayer geometries, the analysis included Butler matrices with open stubs, modified hybrid couplers, no crossovers, no phase shifters, and metamaterial transmission lines. Similarly, tri-layer Butler matrices with metal–ground–metal and metal–metal–ground layers were reviewed. For the quad-layer Butler matrices, ground–metal–ground–metal implementations were investigated in the analysis. The analysis concluded that the bi-layer metamaterial Butler matrix design configuration exhibits lower insertion loss, lower phase error, compact design, excellent bandwidth, good S-parameter performance, and can be fabricated at a lower cost.

A substrate integrated waveguide (SIW)-based 4 × 4 Butler matrix operating at 77 GHz has been reported in [19]. Investigation shows that both of these advanced techniques require complex design approaches to direct the propagation of electromagnetic signals to realize the beamforming and beam steering capabilities. However, the fabrication of the metamaterial-based transmission lines and substrate-integrated waveguides is complex and expensive.

Investigations by the authors showed that as the microstrip transmission lines are comparatively easier to design and fabricate, the design and fabrication complexity of metamaterial or SIW-based Butler matrices can be minimized by realizing appropriate-geometry microstrip transmission lines to realize the Butler matrix functional blocks, along with the necessary input and output ports. Due to easier small-form-factor fabrication of such microstrip Butler matrices, they can easily be integrated to realize smaller in form factor but superior in functionality automotive radars.

In this context, this paper presents the design and ADS simulation results of an HPFS glass-substrate-based 4 × 4 and 8 × 8 microstrip Butler matrices operating at 77 GHz. A fabrication method developed in consultation with The Interdisciplinary Institute for Technological Innovation (3IT) of the Université de Sherbrooke to fabricate the device is also presented. The rest of the paper has been organized in the following manner: Section 1 describes the state-of-the-art Butler matrices and their applications to realize high performance automotive radars. In Section 2, the theory and design challenges of realizing high performance W-band Butler matrices are described. Section 3 presents the design and ADS simulation results of the microstrip building blocks and the target 4 × 4 and 8 × 8 Butler matrices. Section 4 compares the simulated performance parameters with the published results elsewhere to validate the design process. Section 5 presents a fabrication method to fabricate the devices. Finally, Section 6 provides the concluding remarks.

## 2. Butler Matrix Theory and Design Challenges

The Butler matrix is a passive beamforming network having *N* = 2*^n^* input ports (also known as beam ports) and *N* = 2*^n^* output ports, where *n* is a positive non-zero integer. Each input port is electrically connected to all output ports while ensuring high isolation among the input ports. During operation, an RF signal is fed to one of the input ports. The corresponding signals received at the output ports are fed to an antenna array such that the phase difference among the antenna array elements remains the same.

### 2.1. Working Principle of a 4 × 4 Butler Matrix

For a 4 × 4 Butler Matrix, *n* = 2, which leads to *N* = 4 to generate four output beam patterns. The block diagram of a 4 × 4 Butler matrix is shown in Figure 3. The device comprises four input ports, four output ports, 90° hybrid couplers, crossovers, and 45° phase shifters, as shown in Figure 3.

When the four output ports of the device are connected to four linear antenna arrays, four beam patterns are generated, whose angular orientations depend on their respective excitation beam ports. The phase difference $\phi_{p}$ (in degrees) between the RF signals at the adjacent output ports can be calculated from
(1)$$\phi_{p}=\pm \frac{2p-1}{N}\times 180\text{°}$$
where N=4, p=1,2,…,(n+1),n=2 for a 4 × 4 Butler matrix.

One could easily trace the expected output phase by following the path between respective input and output ports and then calculating the expected phase difference between adjacent output ports. The respective beam angle *θ_p_* can be calculated following [18]
(2)$$\sin\theta_{p}=\frac{\lambda }{d}\times \frac{\phi_{p}}{360{}^{\circ}}$$
where λ is the wavelength of the RF signal and *d* is the distance between the feed points of two antenna elements.

Following [11], the antenna feed points must be spaced at λ/2 to have proper beam shape with nulls at integer multiples of λ/2. Accordingly, the phase distribution and respective beam patterns of a 4 × 4 Butler matrix are provided in Table 2. If the phase difference $\phi_{p}$ is obtained as per Table 2, then beam patterns could definitely be seen at the respective beam angle $\theta_{p}$. Hence, for designing a Butler matrix, every component must be designed in such way that it contributes to maintaining the expected phase difference across adjacent output ports.

### 2.2. Working Principle of an 8 × 8 Butler Matrix

An 8 × 8 Butler matrix comprises twelve 90° hybrid couplers, sixteen 0 dB crossovers, and eight phase shifters to provide the necessary phase shifts across the output ports. They are usually connected as shown in Figure 4.

The phase distributions in an 8 × 8 Butler matrix calculated following (1) and (2) are provided in Table 3. Following (1) and (2), for an 8 × 8 Butler matrix, *n* = 3, *N* = 8, and *d* = λ/2. Thus, the design challenge is to determine the optimized dimensions and performance parameters of the 90° hybrid couplers, 0 dB crossovers, and phase shifters to achieve the output beam patterns at the beam angle *θ_p_*, as listed in Table 3.

## 3. Design and Simulation of 4 × 4 and 8 × 8 Microstrip Butler Matrices

### 3.1. Material Selection

A typical microstrip transmission line geometry comprises a slab of dielectric substrate sandwiched between a metallic conducting layer and a metallic ground plane. The authors of [20,21] concluded that ultra-thin high purity glass wafers or thin films are better suited as the dielectric material for W-band microstrip transmission lines due to their superior performance parameters compared to organic and ceramic substrates in terms of reliability, surface roughness, loss tangent, thickness, and dimensional and thermal stability. Accordingly, an HPFS glass substrate from Corning™ was selected as the dielectric material to realize the microstrip transmission line’s geometry to design the target 77 GHz center frequency, 4 × 4 and 8 × 8 Butler matrices. The selected HPFS glass substrate has a low loss tangent (tan δ) of 0.0005, surface roughness <10 Å, and a dielectric constant (εr) of 3.82 at 77 GHz [22,23,24].

As the dimensions of a microstrip transmission line geometry depend on the properties and thickness of the dielectric substrate, the initial design challenge was to determine the optimized dimensions of the proposed HPFS glass-substrate-based 77 GHz microstrip transmissions lines to realize the 90° hybrid couplers, crossovers, and 45° phase shifters. Two-dimensional and 3D finite element methods (FEM) can be used to design and simulate the building blocks and the complete Butler matrix to optimize their performances and geometric dimensions. Accordingly, an industry-standard electromagnetic simulation tool, ADS from Keysight technologies™, was used to conduct the simulations.

Simulation studies conducted in ADS revealed that the width of the microstrip transmission line at the input ports needs to be within 0.1–0.2 mm to excite the 77 GHz RF signals into the Butler matrix network with minimum return loss. It is also necessary to configure all ports as TML ports (Transmission line port calibration in ADS) to match the characteristic impedance of all the components of the Butler Matrix at 77 GHz. The study also revealed that the characteristic impedance of the microstrip lines in the target Butler matrix needs to be within 70 to 100 Ω. Following the ADS guidelines, the substrate’s lateral extension was set to 1.18 mm, and substrate vertical extension was set to 3 mm. The value of 1.18 mm is as per the characteristic impedance of 100 ohms and effective electrical length of $\lambda_{g}/2$ at 77 GHz calculated in CILD (Controlled Impedance Line Designer) in ADS, and 3 mm was selected, as it was greater than 10 times the substrate thickness (*H*). The substrate wall boundary was selected as open to enable EM signal radiation in air. A permanent conductor or permanent magnetic boundary was not used to avoid zero electric field or zero magnetic field at the boundary, contributing to no radiation. With open boundary conditions, EM currents and radiation can be computed by ADS EM solvers easily.

### 3.2. 90° Hybrid Coupler

The 90° hybrid coupler is a four-port directional coupler which is used to divide the input power equally at respective output ports of the coupler and provide a 90° phase difference across the output ports. Figure 5 shows the layout of the designed 77 GHz 90° hybrid coupler in ADS. The corresponding widths and lengths were determined by repetitive parametric optimization techniques in ADS and are listed in Table 4.

The 3D FEM simulation models of the 90° hybrid coupler at 77 GHz are shown in Figure 6, and the simulation results are shown in Figure 7. As can be seen in Figure 7, the phase shift between the output signals at port 2 and 3 of the 90° hybrid coupler was 89.812 degrees.

The simulated insertion loss between port 1 and port 2 is −3.412 dB, and that between port 1 and port 3 is −3.57 dB. The return loss is −34.274 dB, and after isolation of port 1 and port 4 or port 2 and port 3, it is −33.281 dB. The results obtained makes this design configuration of a hybrid coupler suitable for use in 4 × 4 or 8 × 8 Butler matrix or any other type of microstrip device operating at 77 GHz.

### 3.3. Crossover

A crossover is also a directional coupler used to spatially switch a signal in a planar geometry without any coupling and ideally with no loss. To satisfy the design requirements, it was necessary to develop two different types of crossovers, viz., a type 1 and a type 2. Detailed design procedures of both types are provided below.

#### 3.3.1. Type-1 Crossover

A type-1 crossover is a typical linear horizontal orientation crossover without any bends. It can be designed by cascading two 90° hybrid couplers [11,12] such that the signal emerges only at the port diagonal to the input port with theoretically no insertion loss, and there is high isolation among the other ports with theoretically no power output.

However, the initial S-parameter results obtained through ADS simulations did not meet the expectations of 0 dB insertion loss. Several parametric analyses with automatic optimization option in ADS were conducted to determine the optimum geometry of crossover layout, as shown in Figure 8 and Table 5. Corresponding S-parameter values as a function of frequency are shown in Figure 9. The corresponding optimized dimensions of type-1 crossover are given in Table 5.

Figure 9 shows that with the optimized crossover layout dimensions as listed in Table 5, the designed crossover exhibits a 0° phase shift for S_13_ and maintains a high isolation between S_12_ and S_14_ at a return loss S_11_ below −25 dB at 77 GHz. Corresponding insertion loss S_13_ at 77 GHz is −0.636 dB. In terms of power, approx. 93% of power is transmitted from input port to output port and 7% of power is lost between the paths as obtained from ADS.

#### 3.3.2. Type-2 Crossover

A type-2 crossover is a crossover with bends, as shown in Figure 10. This crossover was designed to have a vertical orientation to fit the phase shifters properly in the Butler matrix geometry to reduce the footprint area and maintain high isolation among the other components of the Butler matrix to avoid mutual coupling.

The type-1 crossover shown in Figure 8 was modified accordingly to include the bends that allow for phase adjustment. Several parametric analyses were conducted in ADS to optimize the bend geometries to achieve the target S-parameter values. The simulations revealed that a bend angle of 90 degrees and a curvature radius (*R*) equal to width (*W*) of the microstrip line yields optimum S-parameter values. The optimized type-2 crossover is shown in the Figure 10. The corresponding dimensions of it are given in Table 6.

As can be seen in Figure 11, the return loss of the isolation of port 1 and port 4 is −33.87 dB and that for the isolation port 1 and port 2 is −25.137 dB. The corresponding return loss is −39.614 dB. The insertion loss and phase shift between port 1 and port 3 are −1.081 dB and −0.181°, respectively.

### 3.4. Phase Shifters

Different types of phase shifters were used to adjust the phases of the output signals of the various building blocks in the Butler matrices, as shown in Figure 3 and Figure 4. Microstrip transmission line-type phase shifters were selected, as they are easy to fabricate. Keeping the width of the microstrip line constant and to fit the phase shifters in the networks of 4 × 4 and 8 × 8 Butler matrices appropriately, the lengths *L*_1_, *L*_2_, and *L*_3_ of the phase shifters, as shown in Figure 12, were tuned to obtain correct phases. The horizontal length of the phase shifter, as shown in Figure 12, was kept constant at 2.14 mm to match the horizontal length between port 3 and port 4 in the type-2 crossover, as shown in Figure 10.

As before, parametric analyses in ADS were conducted to optimize the phase shifter dimensions to achieve 22.5°, 45°, and 67.5° phase shifts for the corresponding segments, as shown in Figure 3 and Figure 4. Table 7 shows the final optimized dimensions.

Figure 13a–c shows the 3D FEM simulation results for the designed 22.5°, 45° and 67.5° phase shifters. Similarly, other phase shifters were designed, as shown in the Figure 14, Figure 15, Figure 16 and Figure 17, to adjust the phases of both 4 × 4 and 8 × 8 Butler matrices along with the S-parameters and connect the components to output ports or antenna array appropriately such that the output ports are equally spaced. Equal spacing between the output ports is necessary so that the antenna arrays could be connected properly.

### 3.5. The 4 × 4 Butler Matrix Design

After the successful implementation of the 90° hybrid couplers, crossovers, and phase shifters, these components were integrated as per the block diagrams shown in Figure 3 and Figure 4 to form the target 4 × 4 and 8 × 8 microstrip Butler matrices. Figure 14 shows the layout of the 4 × 4 Butler matrix in ADS.

As is evident, the beam ports P1–P4 in Figure 14 correspond to the beam ports 1L, 2R, 2L, and 1R in Figure 3. Similarly, the output ports P5–P8 in Figure 14 correspond to the ports A1, A2, A3, and A4 in Figure 3. The total footprint area of the completed 4 × 4 Butler matrix is 9.5 mm × 8.3 mm. Figure 15 shows the S-parameters of the designed Butler matrix in ADS obtained through 3D FEM simulations.

The insertion losses between P1 and P5–P8 and P4 and P5–P8 are between −7.5 and −8.8 dB. The insertion losses between P2 and P5–P8 and P3 and P5–P8 are between −7.4 and −10 dB. The return losses at the respective ports are below −20 dB, and the isolation of adjacent input ports was below −20 dB. This 4 × 4 Butler matrix can be connected to a linear microstrip antenna array of four elements to visualize the radiation patterns, as shown in the next section.

### 3.6. The 8 × 8 Butler Matrix

The layout of the complete 8 × 8 Butler matrix is shown in Figure 16. In Figure 16, P1–P8 are the input ports, whereas P9–P16 are the output ports. There are 12 hybrid couplers, 12 crossovers with bends and 4 crossovers without bends, 2 (two) 22.5° phase shifters, 2 (two) 45° phase shifters, 2 (two) 67.5° phase shifters, and a few other phase-adjusting microstrip lines.

A simulation of such a large 8 × 8 Butler matrix is computationally expensive, as approximately 500 GB of virtual memory (or more), along with high speed computing resources, are necessary to compute high accuracy S-parameters and near and far-field electric current distributions. Accordingly, a Linux server with up to 1 TB of memory available in the Research Centre for Integrated Microsystems (RCIM) at the University of Windsor was used. The 192-core Linux server operates at a clock speed of 2.4 GHz. To minimize computation time, ADS simulations were conducted at only three frequencies—76, 77, and 78 GHz. The resulting S-parameter values are provided in Figure 17. Due to the symmetrical structure of the simulated Butler matrix, the output power (dB) results obtained on excitation of ports 1, 2, 3, and 4 (P1–P4) are similar to those for ports 8, 7, 6, and 5 (P8–P5), respectively. From the simulation results in Figure 17, it can be summarized that:

For some cases, the simulated insertion loss is −15 ± 3.5 dB.The insertion losses S_210_ (or S_714_)_,_ S_414_ (or S_515_), and S_415_ (or S_514_) are unexpectedly too high, most likely due to the long propagation path of the RF signal.The return losses for all the ports are less than −20 dB.The isolation between adjacent ports is below −20 dB for all the ports except S_14_ (or S_85_).The phase errors at the output ports are within ±15° compared to phases mentioned in Table 3.

This 8 × 8 Butler matrix can be connected across a microstrip antenna array to view the eight beam patterns generated from individual excitations of the input ports.

## 4. Design Validation

### 4.1. Microstrip Patch Antenna Design

To validate the designed microstrip Butler matrices, two microstrip antenna arrays (one 4 × 1 and one 8 × 1) were designed, and the Butler matrices were simulated in ADS by connecting them to the designed antenna arrays to observe the beam shapes and steered radiation beam patterns. The 4 × 1 and 8 × 1 antenna arrays were designed using inset-fed [16,24] microstrip patch antennas with HPFS glass substrates as the dielectric material. The properties of the HPFS glass substrate remained the same as those used for the Butler matrices.

The ADS optimized geometry and return loss of one of the inset-fed patch antennas in the arrays, as optimized through parametric simulation studies in ADS, are shown in Figure 18, which shows a very low return loss of −30.478 dB at 77 GHz. The corresponding dimensions are given in Table 8.

The beam pattern of the designed microstrip patch antenna, as obtained from ADS simulations, is shown in Figure 19. The green markers in Figure 19a,b show the plane of the azimuth angle (φ) for which the radiation patterns were calculated. A comparison of 2D radiation patterns plotted in rectangular coordinates, as shown in Figure 19c,d, shows that at φ = 90°, the gain is slightly higher than at φ = 0°. However, the elevation angle (*θ*) is 0° for φ = 0°, and it is 12° for φ = 90°.

### 4.2. The 4 × 4 Butler Matrix with an Antenna Array of Four Elements

Both the 4 × 4 and 8 × 8 Butler matrices were then simulated by connecting them to suitably sized antenna arrays to obtain the beam shapes and steered beam patterns.

Figure 19 shows the simulation model of the 4 × 4 Butler matrix after connecting to the 4 × 1 antenna array. The complete structure was then simulated in a 3D FEM environment in ADS. The resulting radiation patterns are shown in Figure 20.

As it is evident in Figure 21, selective excitation of input ports 1, 2, 3, and 4 steers the beam axes to *θ* = −15°, 42°, −42°, and 15°, respectively, in a 90° azimuthal plane (φ = 90°). The maximum gain of 20.322 dB was observed for port 4′s excitation, and the gain was minimal at 16.763 dB when port 2 was excited. The difference between the main lobe’s gain and side lobe’s gain is higher than 14 dB when port 1 (or 1L) or port 4 (or 1R) is excited, but lower than 10 dB when port 2 (or 2R) or port 3 (or 2L) is excited. All four beam patterns can be viewed in one graph with the help of a ‘History’ option available in the ADS data plot. The ‘H’ symbol depicts that the ‘History’ option is on. Table 9 compares the calculated beam angles, as mentioned in Table 1, and observed beam angles. The highest beam angle error was observed to be within the ±6.6° limits.

### 4.3. The 8 × 8 Butler Matrix with an Antenna Array of Eight Elements

Figure 22 shows the simulation model of the 8 × 8 Butler matrix after connecting to the 8 × 1 antenna array. The complete structure was then simulated in 3D FEM environment in ADS. The resulting radiation patterns are shown in Figure 23 for individual excitation of the beam ports.

### 4.4. Performance Comparison of 4 × 4 and 8 × 8 Butler Matrices at 77 GHz

Table 10 summarizes the key radiation beam pattern simulation results for the 4 × 4 and 8 × 8 Butler matrices at 77 GHz.

Table 11 compares the performance parameters of the designed 4 × 4 Butler matrix in this paper with those of the 4 × 4 Butler matrix presented in [19]. As it is evident, the 4 × 4 Butler matrix presented in this paper offers lower insertion loss over a much smaller footprint area and less thickly, along with comparable results for other parameters. The significant footprint area advantage (approximately 10 times smaller) of the new design over the design presented in [19] will enable others to design much smaller automotive radars to achieve similar or superior beamforming characteristics. To the best of the authors’ knowledge, there are no 77 GHz, 8 × 8 Butler matrix performance results available in the literature. It was not possible to compare the performance results of the 8 × 8 Butler matrix. However, as the building blocks of both the 4 × 4 and 8 × 8 Butler matrices are the same, it is expected that the 8 × 8 Butler matrix will also be able to provide superior performance over a small footprint area, contributing to smaller automotive radars with superior beamforming characteristics.

## 5. Butler Matrix Fabrication

A microfabrication technique to fabricate the designed Butler matrices has been developed in consultation with The Interdisciplinary Institute for Technological Innovation (3IT) of the Université de Sherbrooke in Sherbooke, Quebec, Canada. The technique involves the realization of patterned gold thin films on a 200 μm thick A Corning HPFS glass substrate using a lift-off process realizes the microstrip lines. The major fabrication steps are described below.

A 200 µm thick HPFS glass substrate from Corning is RCA cleaned. A 0.8 µm thick layer of gold is deposited by an e-beam evaporation method onto the backside of the wafer using a 30 nm thick titanium adhesion layer (Figure 24a). An LOR 30B lift-off resist was then spin-deposited at 2000 rpm and baked at 170 °C for 10 min, as shown in Figure 24b. A thin film of Shipley 1827 photoresist is then spin deposited at 2500 rpm and baked for 3 min at 113 °C (Figure 24c). The LOR and Shipley 1827 layers are then patterned by exposing the resist in a Karl Suss MA6 aligner with 540 mJ/cm^2^ and developed for 3 min in a MF-319 dish (Figure 24d). Following a de-ionized (DI) water rinse in the spin rinse dryer, a 0.8 µm thick layer of gold is deposited by an e-beam evaporation method (Figure 24e), on the topside of the processed wafer. The resist is then dissolved in Remover PG during an overnight soak to complete the metal lift-off step (Figure 24f). The wafer is then bonded to a carrier wafer with a layer of crystal bond to provide mechanical support during subsequent dicing to complete the fabrication process. The developed process will be implemented, and the devices will be tested once the optimization process is completed. The measurement results will be published in a future research paper.

## 6. Conclusions

The designed 77 GHz microstrip 4 × 4 and 8 × 8 Butler matrices on a 200 μm thick HPFS glass substrate can provide superior beamforming performance at a lower cost, with a smaller size, and with less thickness to realize compact radars to improve road safety and driving comfort for vehicles with advanced driver-assistance systems (ADAS) and autonomous vehicles. The conducted simulation studies revealed that the glass substrate from Corning™ has superior loss characteristics to minimize insertion loss at high frequencies. However, the optimization of the phase characteristics and insertion losses in a 3D FEM simulation environment requires more than 500 GB of memory. The time taken by ADS solvers to run one simulation is computationally intensive. Further optimization is necessary to improve the insertion loss and the phase error. A microfabrication technique using a standard lift-off process has been developed in consultation with The Interdisciplinary Institute for Technological Innovation (3IT) of the Université de Sherbrooke in Sherbooke, QC, Canada. The device will be fabricated and tested once the optimization process is completed.

## Figures and Tables

**Figure 1 sensors-23-01418-f001:**
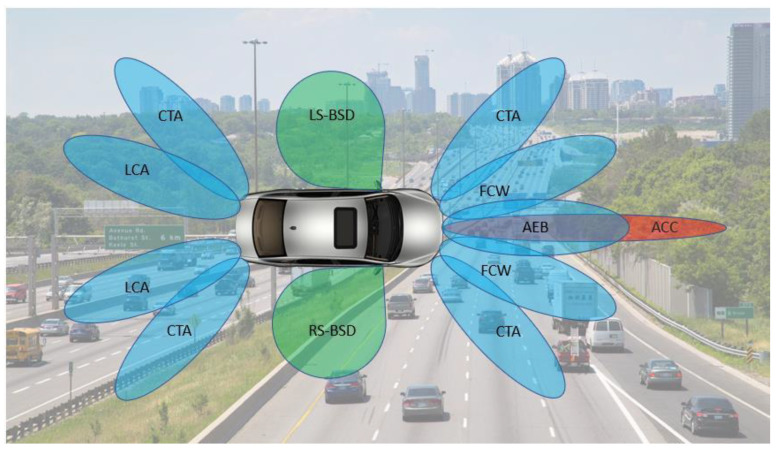
Applications of radars in vehicles for adaptive cruise control (ACC), automatic emergency braking (AEB), left side (LS) and right side (RS) blind spot detection (BSD), collision mitigation (CM), cross traffic alerts (CTAs), forward collision warning (FCW), and lane change assistants (LCAs).

**Figure 2 sensors-23-01418-f002:**
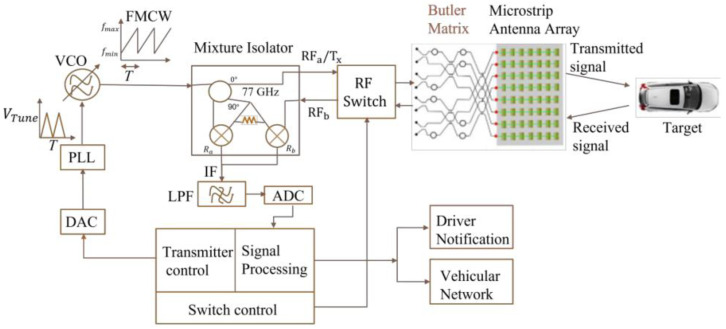
Block diagram of a 77 GHz automotive radar with an 8 × 8 Butler matrix.

**Figure 3 sensors-23-01418-f003:**
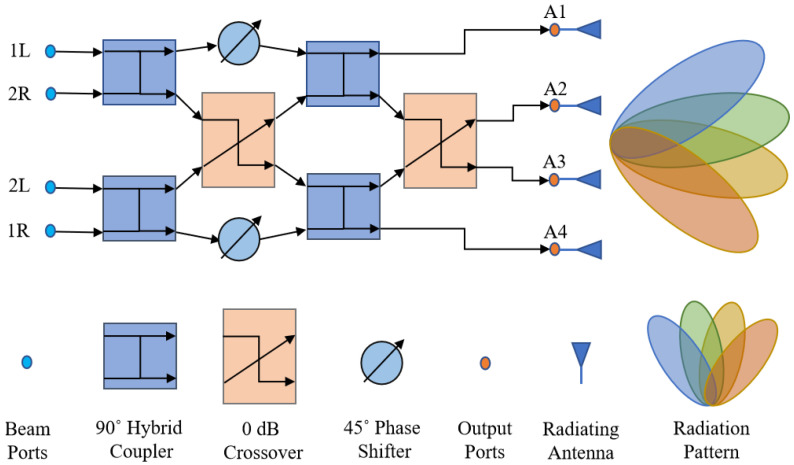
Operational block diagram of 4 × 4 Butler matrix.

**Figure 4 sensors-23-01418-f004:**
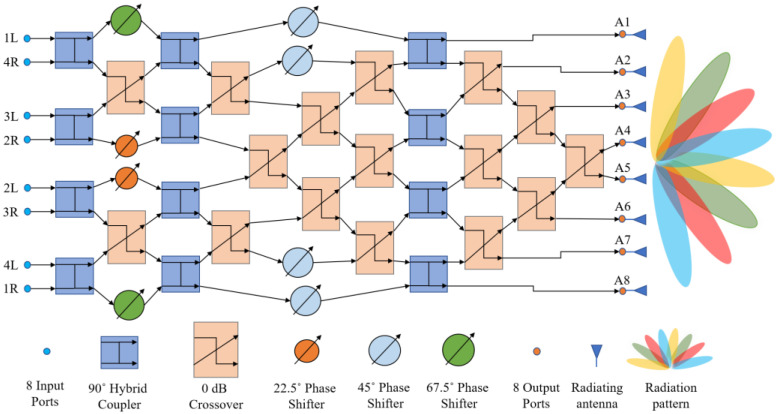
Operational block diagram of an 8 × 8 Butler matrix.

**Figure 5 sensors-23-01418-f005:**
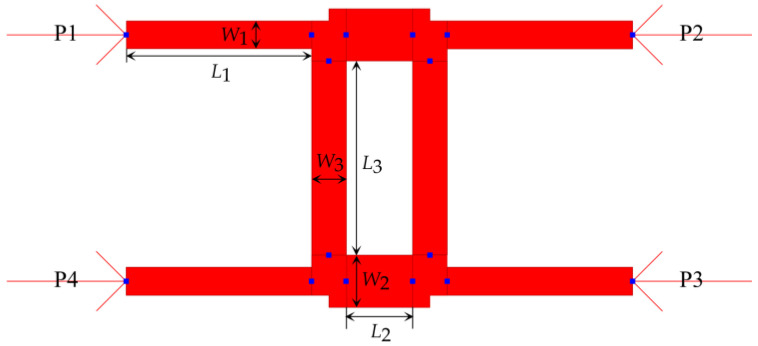
Layout of the 77 GHz microstrip 90° hybrid coupler in ADS.

**Figure 6 sensors-23-01418-f006:**
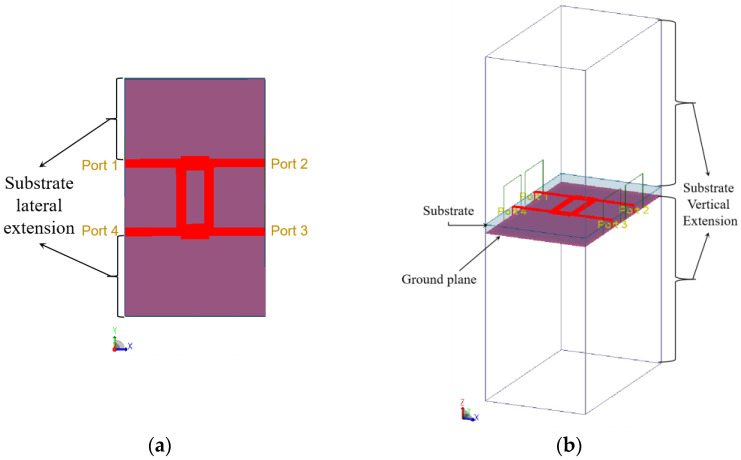
(**a**) Topview and (**b**) isometric view of the 3D FEM simulation model of the 90° hybrid coupler in ADS.

**Figure 7 sensors-23-01418-f007:**
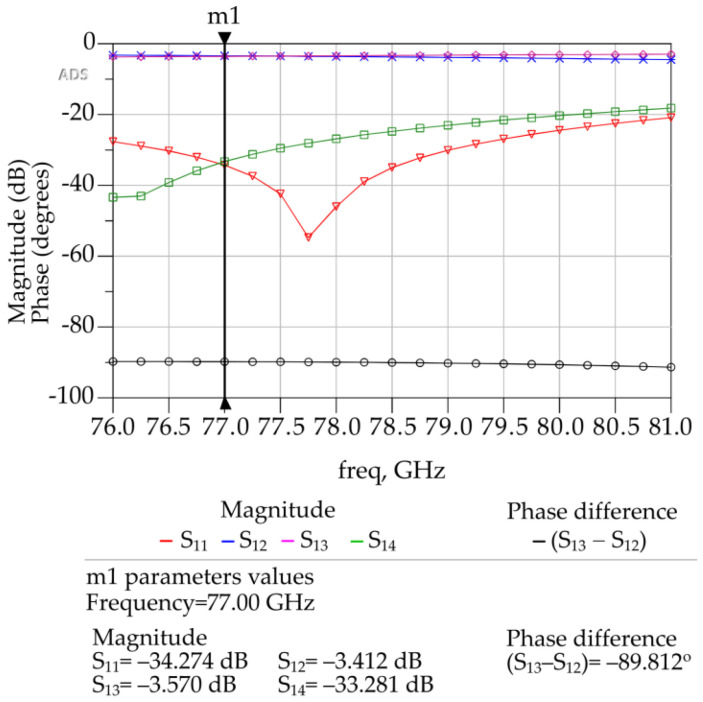
ADS generated 3D FEM simulation results for the designed 90° hybrid coupler.

**Figure 8 sensors-23-01418-f008:**
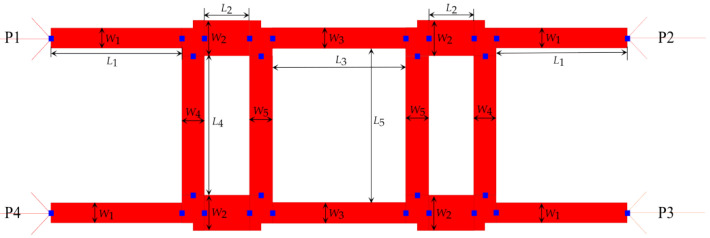
Layout of the type-1 crossover in ADS.

**Figure 9 sensors-23-01418-f009:**
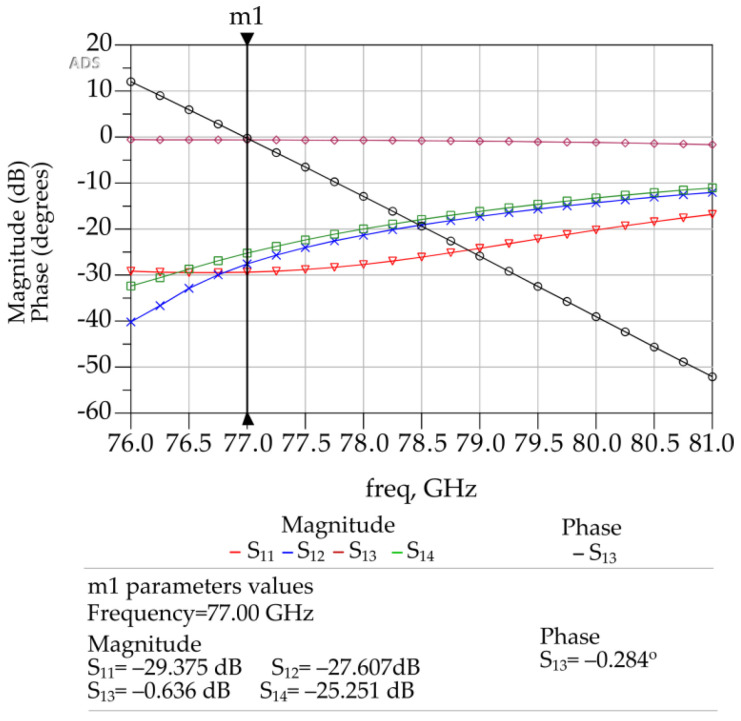
ADS generated 3D FEM simulation results for the type-1 crossover.

**Figure 10 sensors-23-01418-f010:**
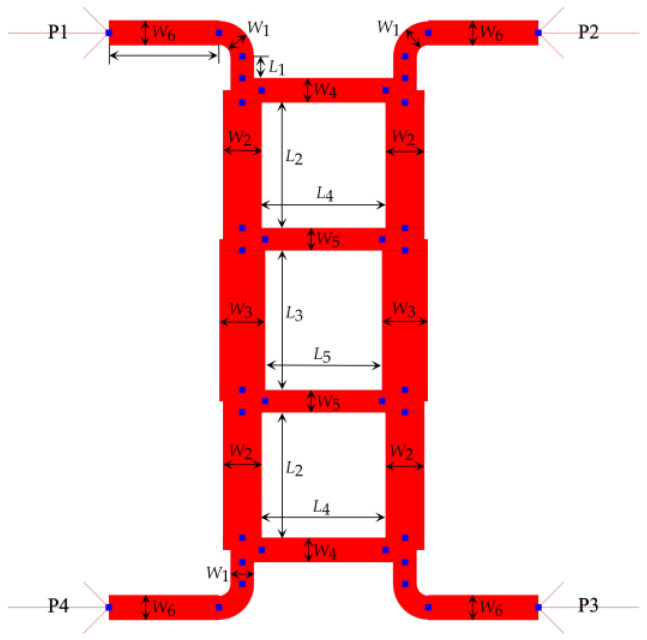
Layout of the type-2 crossover in ADS.

**Figure 11 sensors-23-01418-f011:**
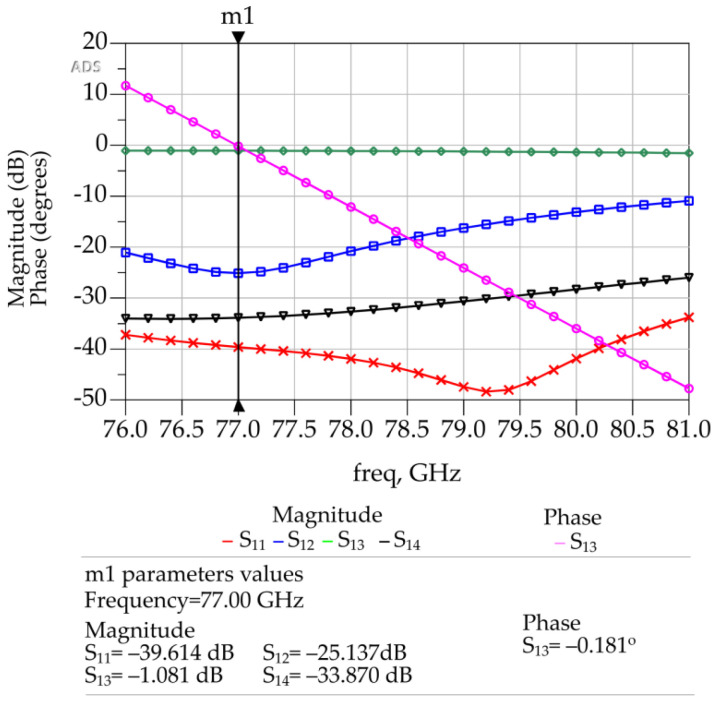
ADS generated 3D FEM simulation results of the type-2 crossover.

**Figure 12 sensors-23-01418-f012:**
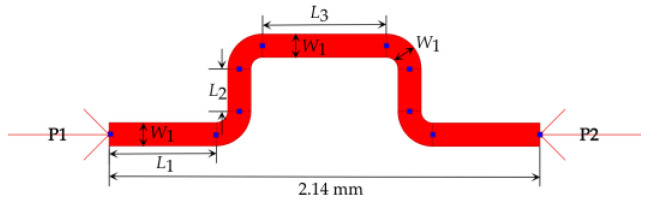
Layout of a microstrip phase shifter in ADS.

**Figure 13 sensors-23-01418-f013:**
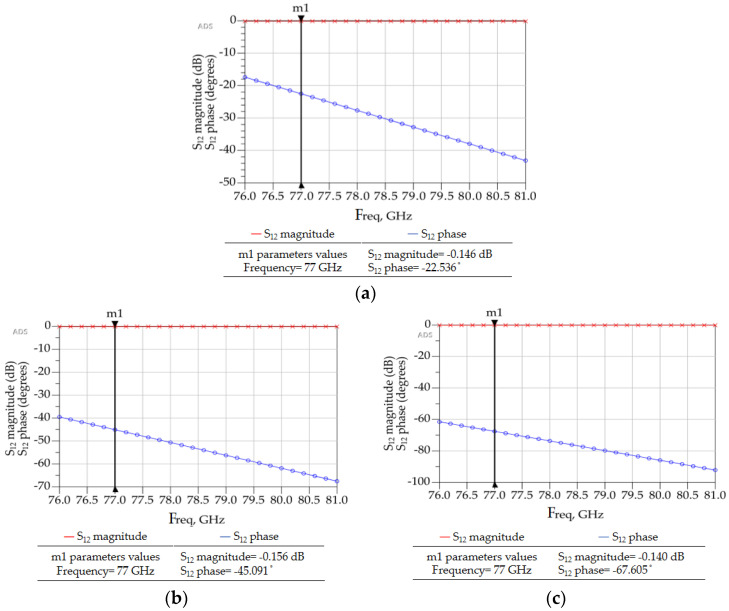
ADS generated 3D FEM simulation results for (**a**) a 22° phase shifter, (**b**) a 45° phase shifter, and (**c**) a 67.5° phase shifter.

**Figure 14 sensors-23-01418-f014:**
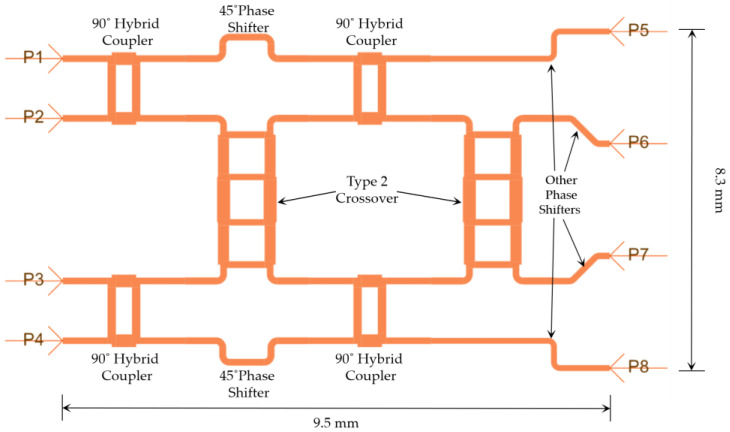
Layout of the 4 × 4 Butler matrix in ADS.

**Figure 15 sensors-23-01418-f015:**
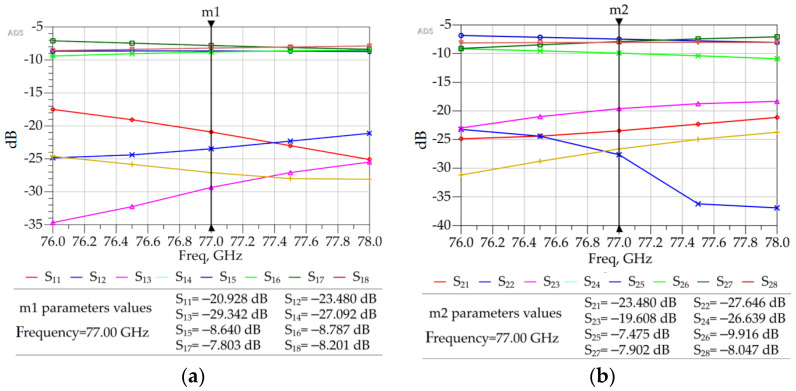
ADS generated 3D FEM simulation results for the 4 × 4 Butler matrix. (**a**) Isolation, return loss, and insertion losses with port P1 or P4 excited. (**b**) Isolation, return loss, and insertion losses with port P2 or P3 excited.

**Figure 16 sensors-23-01418-f016:**
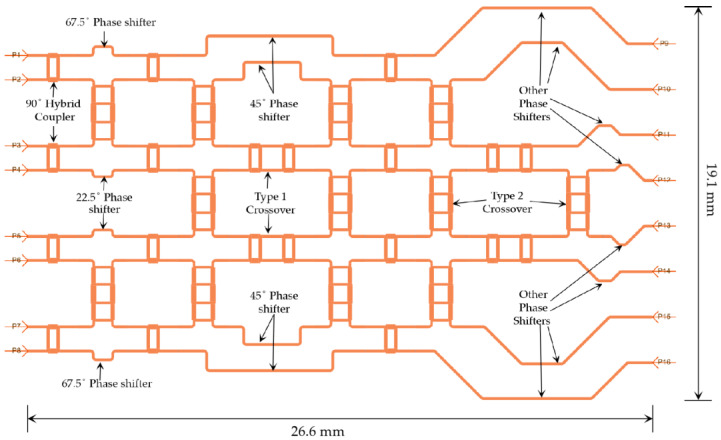
Layout of the 8 × 8 Butler matrix in ADS.

**Figure 17 sensors-23-01418-f017:**
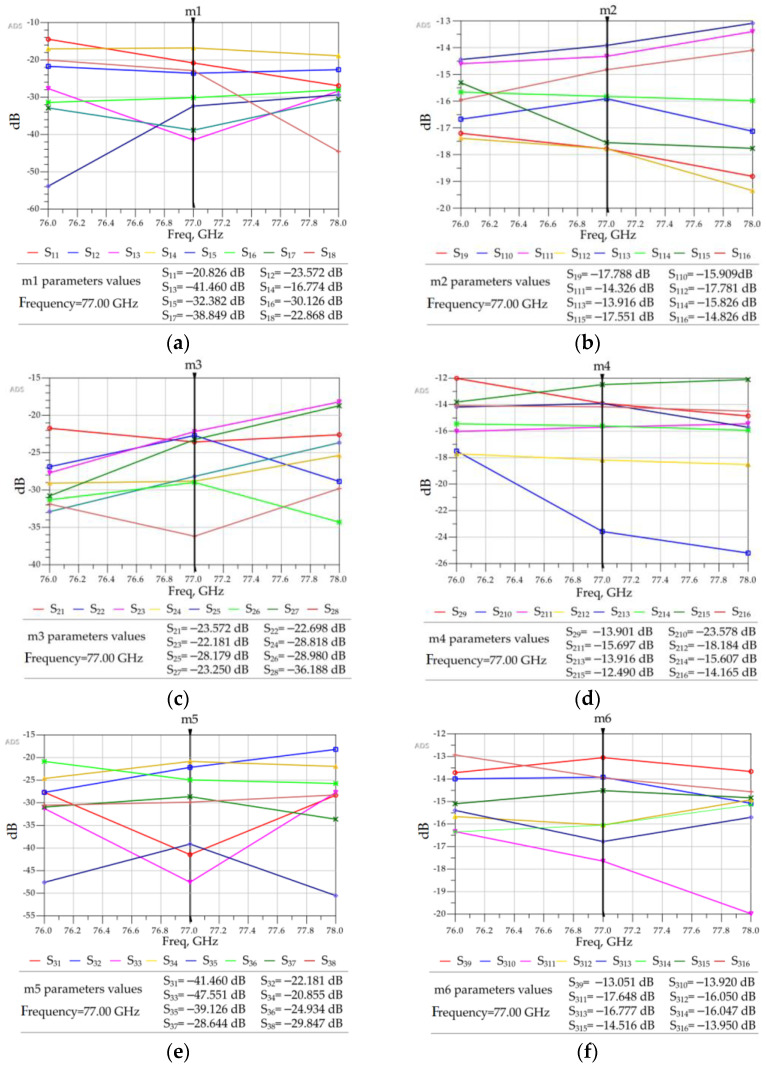
ADS generated 3D FEM simulation results for the 8 × 8 Butler matrix. (**a**) Return loss and isolation between input ports when P1 or P8 is excited. (**b**) Insertion losses between P1 or P8 and eight output ports, P9–P16, when port P1 or P8 is excited. (**c**) Return loss and isolation between input ports when P2 or P7 is excited. (**d**) Insertion losses between P2 or P7 and eight output ports when P2 or P7 is excited. (**e**) Return loss and isolation between input ports when P3 or P6 is excited. (**f**) Insertion losses between port P3 or P6 and eight output ports when P3 or P6 is excited. (**g**) Return loss and isolation between input ports when P4 or P5 is excited. (**h**) Insertion losses between port P4 or P5 and eight output ports when P4 or P5 is excited.

**Figure 18 sensors-23-01418-f018:**
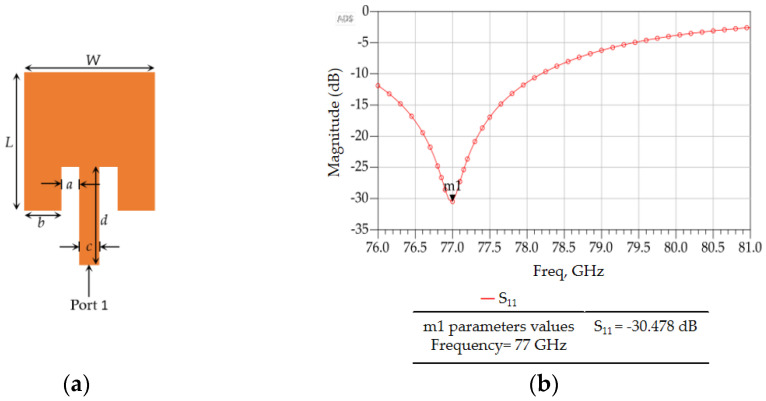
(**a**) Layout and (**b**) return loss of the designed inset–fed microstrip patch antenna.

**Figure 19 sensors-23-01418-f019:**
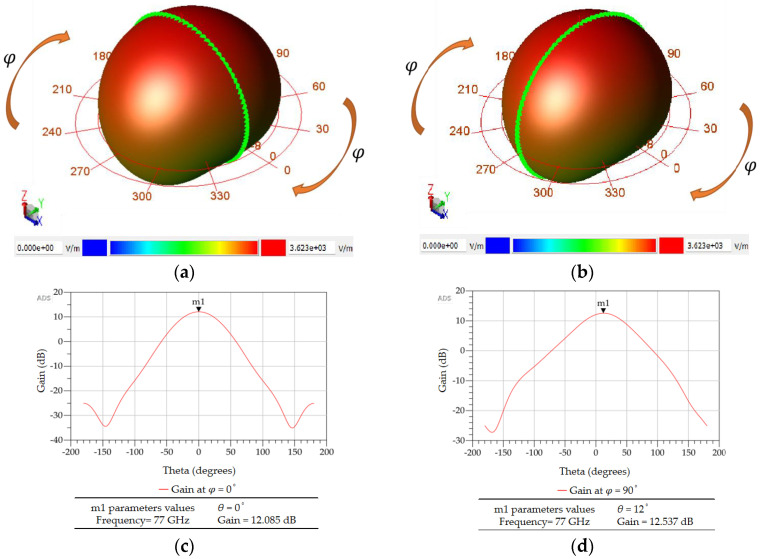
(**a**) 3D radiation pattern of a single inset–fed microstrip patch antenna at φ = 0°. (**b**) 3D radiation pattern of the same antenna at φ = 90°. (**c**) 2D radiation pattern plotted in rectangular coordinates at φ = 0°. (**d**) 2D radiation pattern plotted in rectangular coordinates at φ = 90°.

**Figure 20 sensors-23-01418-f020:**
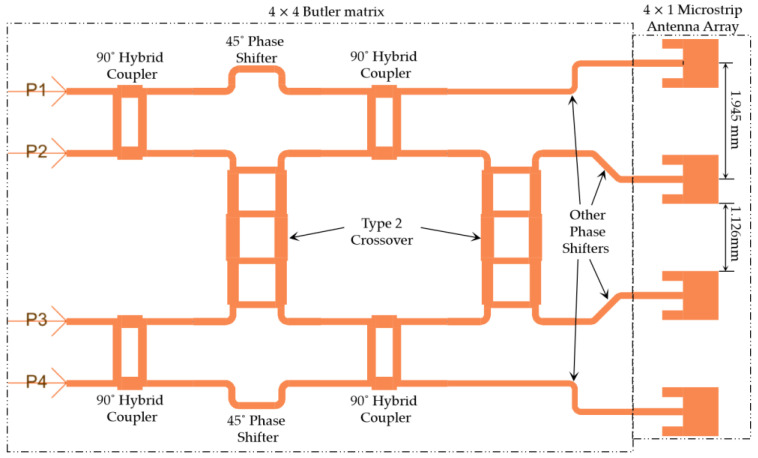
A 4 × 4 Butler matrix with a microstrip antenna array of four elements in ADS.

**Figure 21 sensors-23-01418-f021:**
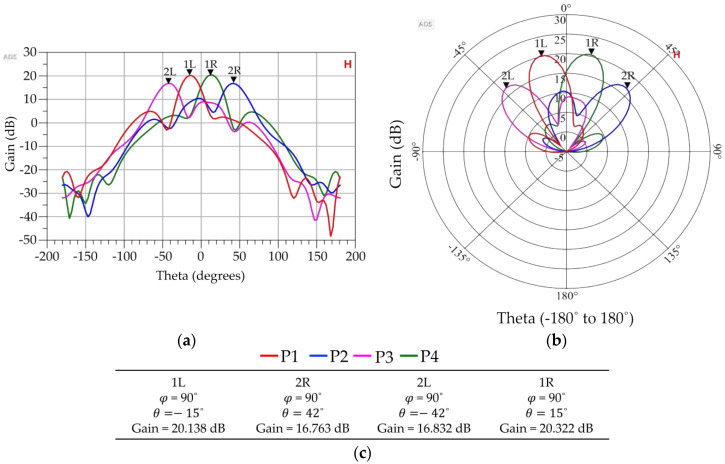
Three dimensional FEM simulated radiation patterns generated by ADS. (**a**) Radiation patterns in rectangular coordinates when beam ports 1L, 2R, 2L, and 1R are excited individually, (**b**) Radiation patterns in polar coordinates when beam ports 1L, 2R, 2L, and 1R are excited individually, (**c**) Corresponding parameters values of respective beam ports.

**Figure 22 sensors-23-01418-f022:**
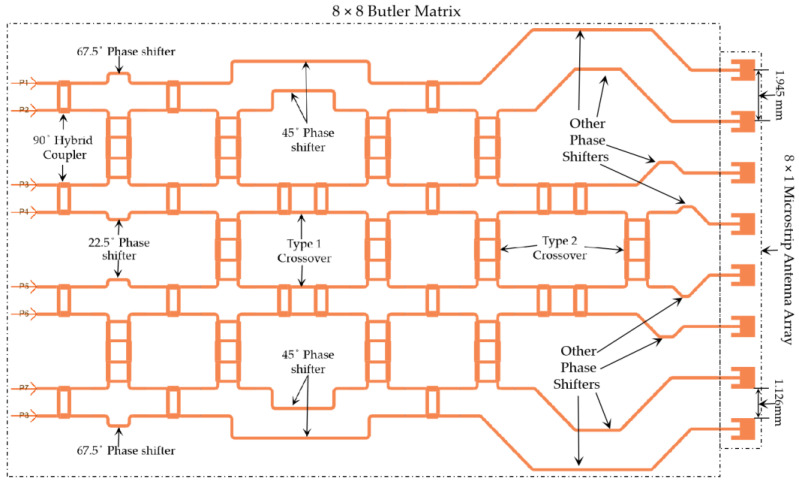
An 8 × 8 Butler matrix with a microstrip antenna array of eight elements in ADS.

**Figure 23 sensors-23-01418-f023:**
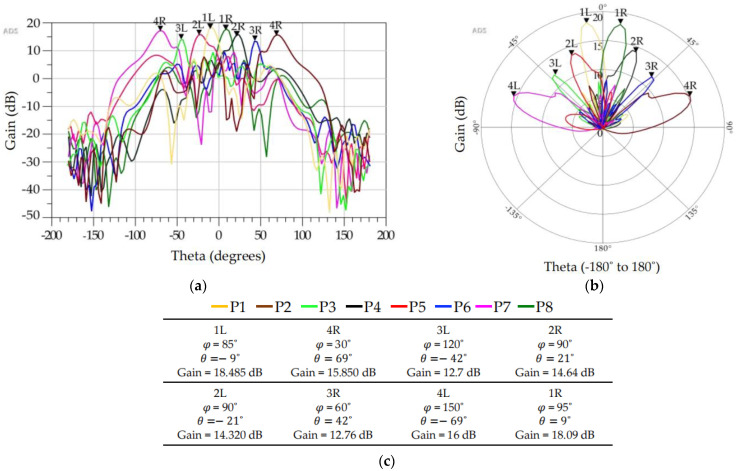
Three dimensional FEM simulated radiation patterns generated by ADS. (**a**) Radiation patterns in rectangular coordinates when the beam ports are excited individually, (**b**) radiation patterns in polar coordinates when beam ports are excited individually, (**c**) corresponding simulation parameters for the beam ports.

**Figure 24 sensors-23-01418-f024:**
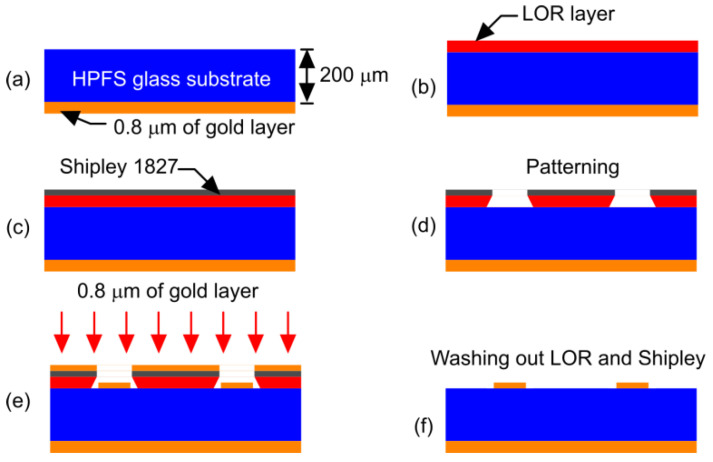
A tentative fabrication method to fabricate the designed Butler matrices. (**a**) Deposition of titanium layer and gold layer on the backside of the RCA cleaned HPFS glass substrate layer. (**b**) Deposition of LOR layer on the top layer of HPFS glass substrate. (**c**) Deposition of photoresist layer over the LOR layer. (**d**) Patterning of LOR and photoresist layers. (**e**) Deposition of gold layer by e-beam evaporation method. (**f**) Removal the LOR and photoresist layers to get the final Butler matrix circuit patterned with gold layer.

**Table 1 sensors-23-01418-t001:** Typical specifications of automotive radars [2].

Radar Type	Frequency (GHz)	Bandwidth (GHz)	Angle of Coverage	Range (m)	Resolution (m)
SRR	77–81	4	±20–50°	0.15–30	~0.1
MRR	77–81	4	±6–10°	0.2–100	~0.5
LRR	76–77	1	±5–7.5°	10–250	~0.5

**Table 2 sensors-23-01418-t002:** Phase distribution in a 4 × 4 Butler matrix.

Output Ports	Beam Ports			
	1L	2R	2L	1R
A1	−45°	−135°	−90°	−180°
A2	−90°	0/360°	−225°/135°	−135°
A3	−135°	−225°/135°	0°/360°	−90°
A4	−180°	−90°	−135°	−45°
Phase difference ($\phi_{p}$)	45°	−135°	135°	−45°
Beam angle $\left(\theta_{p}\right)$	−14.47°	48.6°	−48.6°	14.47°

**Table 3 sensors-23-01418-t003:** Phase distribution in an 8 × 8 Butler matrix.

	Beam Ports							
Output Ports	1L	4R	3L	2R	2L	3R	4L	1R
A1	−112.5°	157.5°	−135°	135°	−112.5°	157.5°	−180°	90°
A2	−135°	−45°	112.5°	−157.5°	−180°	−90°	22.5°	112.5°
A3	−157.5°	112.5°	0°	−90°	112.5°	22.5°	−135°	135°
A4	−180°	−90°	−112.5°	−22.5°	45°	135°	67.5°	157.5°
A5	157.5°	67.5°	135°	45°	−22.5°	−112.5°	−90°	−180°
A6	135°	−135°	22.5°	112.5°	−90°	0°	112.5°	−157.5°
A7	112.5°	22.5°	−90°	−180°	−157.5°	112.5°	−45°	−135°
A8	90°	−180°	157.5°	−112.5°	135°	−135°	157.5°	−112.5°
Phase difference ($\phi_{p}$)	22.5°	−157.5°	112.5°	−67.5°	67.5°	−112.5°	157.5°	−22.5°
Beam angle $\left(\theta_{p}\right)$	−7°	61°	−39°	22°	−22°	39°	−61°	7°

**Table 4 sensors-23-01418-t004:** Dimensions of the microstrip 90° hybrid coupler.

Parameter	*W* _1_	*L* _1_	*W* _2_	*L* _2_	*W* _3_	*L* _3_
Values (mm)	0.118	0.779	0.145	0.816	0.22	0.28

**Table 5 sensors-23-01418-t005:** Optimized dimensions of the type-1 crossover.

Parameters	*W* _1_	*L* _1_	*W* _2_	*L* _2_	*W* _3_	*L* _3_	*W* _4_	*L* _4_	*W* _5_	*L* _5_
Values (mm)	0.119	0.834	0.211	0.29	0.124	0.855	0.12	0.825	0.11	0.912

**Table 6 sensors-23-01418-t006:** Optimized dimensions of the type-2 crossover.

Parameters	*W* _1_	*W* _2_	*W* _3_	*W* _4_	*W* _5_	*W* _6_
Values (mm)	0.116	0.192	0.226	0.12	0.11	0.122
**Parameters**	** *L* _1_ **	** *L* _2_ **	** *L* _3_ **	** *L* _4_ **	** *L* _5_ **	** *L* _6_ **
Values (mm)	0.107	0.618	0.686	0.616	0.582	0.547

**Table 7 sensors-23-01418-t007:** Optimized dimensions of the phase shifters.

		Parameters			
		*W* _1_	*L* _1_	*L* _2_	*L* _3_
Values (mm)	22.5°	0.119	0.532	0.0533	0.616
	45°	0.119	0.532	0.136	0.616
	67.5°	0.119	0.532	0.2025	0.616

**Table 8 sensors-23-01418-t008:** Inset-fed microstrip patch antenna dimensions.

Parameters	*W*	*L*	a	b	c	d
Values(mm)	0.819	0.9453	0.187	0.163	0.119	0.8434

**Table 9 sensors-23-01418-t009:** Calculated and observed beam angles for the 4 × 4 Butler matrix after connecting to the 4 × 1 antenna array for individual beam port excitation.

	Beam Ports			
Beam Angle (*θ_p_*)	1L	2R	2L	1R
Calculated	−14.47°	48.6°	−48.6°	14.47°
Observed	−15°	42°	−42°	15°
Error	−0.53°	−6.6°	6.6°	0.53°

**Table 10 sensors-23-01418-t010:** Comparison of the performance parameters of the designed 4 × 4 and 8 × 8 Butler matrices.

Parameters	4 × 4 Butler Matrix	8 × 8 Butler Matrix
Return loss	<−7 dB	<−9 dB
Isolation	<−20 dB	<−20 dB
Maximum main lobe	10.16 dBi	9.1 dBi
Minimum main lobe	9 dBi	6.4 dBi
Elevation angle error	±6.6°	±8°
Total angular coverage	144°	162°

**Table 11 sensors-23-01418-t011:** Performance comparison of the designed 77 GHz, 4 × 4 Butler matrix.

Parameters	Reference [19]	This Work
Technology	SIW	Microstrip
Frequency	77 GHz	77 GHz
Simulation software	HFSS	ADS
Matrix type	4 × 4	4 × 4
Substrate	RT/Duroid 6002	HPFS
Thickness of substrate	0.508 mm	0.2 mm
Footprint area	31.5 × 28.5 mm^2^	9.5 × 8.3 mm^2^
Insertion loss	−6.7 ± 0.75 dB	−8 ± 2 dB
Isolation	<−20 dB	<−20 dB
Footprint area of antenna array	9 × 8.4 mm^2^	2.61 × 8.994 mm^2^
Antenna type	Slot	Microstrip
Return loss	<−10 dB	<−7 dB
Maximum main lobe power	12.21 dBi	10.16 dBi
Minimum main lobe power	9.9 dBi	9 dBi
Phase error	7°	6.6°

## Data Availability

Data sharing not available.

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
