# Peer review of "A Low-Loss, 77 GHz, 8 × 8 Microstrip Butler Matrix on a High-Purity Fused-Silica (HPFS) Glass Substrate"

_sensors, 2023, doi:10.3390/s23031418_

Round 1

Reviewer 1 Report

1) On what basis High Purity Fused Silica (HPFS) glass substrate is selected?

2) A comparison table is needed to prove the novelty of this work with the state of art work done previously.

3)What the variables and/or constants in the related equation represent should be stated after the equation.

4) Page 6, in line number 169, 4 x 4 and 8 x 8, "4" is missing there.

5) Fig. 5,8,10,13, change the color for the better visibility and readability.

6)The figures from ADS has high numbers of legends, Please find out an alternative to show it in figure.

7)Is it planned to fabricate this design soon?

8) Manuscript also needed an english grammar expert for improvement in language.

9) How feasible it is to fabricate the device on mm thickness of substrate?

10) Make a comparison using few parameters for 4x4 and 8x8 butler matrix design for same antenna.  

Reviewer 2 Report

Authors are suggested to fabricate the proposed butler matrices with integrated rectangular patch antennas (Figure 21). The fabrication and testing are trivial tasks and are necessary for microstrip components. Without prototyping and testing, the validation of the results is incomplete.

Round 2

Reviewer 1 Report

All the comments given are addressed satisfactorily.

Reviewer 2 Report

The fabrication and measured results are not provided but I believe the work is sufficient for publication. The authors have addressed all other concerns. Now the paper may be accepted in its current form.